# Busting contraception myths and misconceptions among youth in Kwale County, Kenya: results of a digital health randomised control trial

Peter Gichangi,[1,2] Lianne Gonsalves [ID],[3] Jefferson Mwaisaka,[2] Mary Thiongo,[2] Ndema Habib,[3] Michael Waithaka [ID],[2] Tigest Tamrat,[3] Alfred Agwanda,[4] Hellen Sidha,[5] Marleen Temmerman,[6,7] Lale Say[3]

PG and LG contributed equally.

PG and LG are joint first authors.

For numbered affiliations see end of article.

**Correspondence to**
Professor Peter Gichangi; gichangip@yahoo.com

## ABSTRACT

**Objectives** The objective of this randomised controlled trial in Kenya was to assess the effect of delivering sexual and reproductive health (SRH) information via text message to young people on their ability to reject contraception-related myths and misconceptions.

**Design and setting** A three-arm, unblinded randomised controlled trial with a ratio of 1:1:1 in Kwale County, Kenya.

**Participants and interventions** A total of 740 youth aged 18–24 years were randomised. Intervention arm participants could access informational SRH text messages on-demand. Contact arm participants received once weekly texts instructing them to study on an SRH topic on their own. Control arm participants received standard care. The intervention period was 7 weeks.

**Primary outcome** We assessed change myths believed at baseline and endline using an index of 10 contraception-related myths. We assessed change across arms using difference of difference analysis.

**Results** Across arms, <5% of participants did not have any formal education, <10% were living alone, about 50% were single and >80% had never given birth. Between baseline and endline, there was a statistically significant drop in the average absolute number of myths and misconceptions believed by intervention arm (11.1%, 95% CI 17.1% to 5.2%), contact arm (14.4%, 95% CI 20.5% to 8.4%) and control arm (11.3%, 95% CI 17.4% to 5.2%) participants. However, we observed no statistically significant difference in the magnitude of change across arms.

**Conclusions** We are unable to conclusively state that the text message intervention was better than text message 'contact' or no intervention at all. Digital health likely has potential for improving SRH-related outcomes when used as part of multifaceted interventions. Additional studies with physical and geographical separation of different arms is warranted.

**Trial registration number** ISRCTN85156148.

## INTRODUCTION

There is a high unmet need for sexual and reproductive health (SRH) information and services, for both married and unmarried youth worldwide. Data from 61 low-income

## STRENGTHS AND LIMITATIONS OF THIS STUDY

⇒ This study included two digital intervention arms, meaning that it would be possible to determine whether changes in outcomes were due to sexual and reproductive health (SRH) content delivered by phone, or participants being 'nudged' by phone to think about (and learn about) their SRH.

⇒ The study intentionally did not power sample size around SRH behavioural outcomes, building on previous research that light-touch digital interventions alone may not be enough to see behavioural change in such a complex area of health—instead the primary outcome focuses on SRH knowledge.

⇒ A key limitation is that the study's individual-level randomisation of young people living near each other is likely to have resulted in contamination between arms.

and middle-income countries show that 33 million women aged 15–24 have an unmet need for contraception.[1] In Kenya, the 2014 Demographic and Health Survey found that modern contraceptive use among all adolescents age 15–19 years is low (9.3%) compared with all women aged 15–49 years (39.1%).[2] Partly as a result, the number of adolescents aged 15–19 years who were pregnant or mothers has stagnated at 18% between 2008 and 2014.[2 3]

Sexually active young people face a variety of obstacles to access and use modern contraceptives. They may encounter financial, cultural, social, legal barriers, fear of side effects (eg, infertility and adverse reactions) or cultural norms that restrict their access to contraception services in health facilities.[4–7] Additionally, contraception myths and misconceptions can negatively affect access to and use of SRH services.[8 9] Misinformation and myths/misconceptions are often learnt from social networks.[10 11] In this paper,

**BMJ**

we describe myths and misconceptions as those being communal or widespread beliefs about effects of contraceptives, which are distinct from individuals' experiences with contraception-related side effects.[12]

The proliferation of mobile phone technology, and its popularity and ownership with young people in particular,[13][14] provides an innovative way to educate young people on contraception and their health more broadly. There are indications that health promotion campaigns among adolescents and young people through text messaging may contribute to improved SRH knowledge, behaviours and outcomes.[15] However, there is less rigorous research and documentation of SRH mobile phone interventions for adolescents and young people in developing countries.[16] In Kenya in particular, an estimated 93% of households already owned a mobile phone by 2011.[17]

Mobile phone-based digital health interventions have been successfully used in HIV programmes,[18][19] postabortion care[20] and to address chronic disease conditions.[21] Providing broader SRH content, including contraception information, via mobile phones to young people would appear to be a natural strategy to reach them,[22] increase their contraception knowledge[23] and improve correct contraception use.[24] After all, when it comes to 'sensitive' SRH content, mobile phones can privately deliver information without stigma or judgement. However, the evidence that digital health interventions can improve youth SRH-related outcomes, including contraception knowledge and uptake, is yet to be significantly established.[25–27]

To address this gap, the WHO's Department of Sexual and Reproductive Health and Research partnered with research partners in Peru and Kenya to develop the Adolescent/Youth Reproductive Mobile Access and Delivery Initiative for Love and Life Outcomes (ARMADILLO) Study. The ARMADILLO intervention used short message service (SMS, also known as 'text message') to deliver SRH information on-demand via a numbers-based menu. The content was developed in the study's formative stage around several SRH 'domains' of interest to policy-makers and young people alike.[28] The intervention was evaluated using a three-arm randomised controlled trial (RCT). This paper reports on the Kenya study's primary outcome: are young people with access to the ARMADILLO intervention better able to reject contraception myths and misconceptions as compared with periodic SMS encouraging self-learning or usual care (no intervention).

## METHODS
### Study design
This was a three-arm RCT (1:1:1 allocation) involving youth age 18–24 years. The trial ran for 7 weeks, with assessments at baseline and endline. The study methods have been described elsewhere in full,[29] but are described briefly below.

### Participants and setting
The study was conducted in Kwale County, one of the six counties in the Coastal region of Kenya. The study area consisted of select enumeration areas (EAs) in six Kwale County sublocations which border each other: Ngombeni, Kitivo, Simkumbe, Mkoyo, Gombato and Ukunda. Eligibility criteria was as follows: youth (male and female) aged 18–24; literate; had their own mobile phone at the time of recruitment and reported regular use; reported current use of text messaging.

The Kenya National Bureau of Statistics provided a list of EAs for the six sub-locations. From this list, we randomly selected 21 EAs to be mapped. During the mapping (October 2017), data collectors visited all households and enumerated anyone in the home who was age-eligible to participate. Then, we randomly selected one eligible participant from each household. Starting in February 2018, trained data collectors returned to households and attempted to recruit the selected youth. After consenting to participate, participants completed a baseline survey. Both baseline and endline surveys were implemented by trained data collectors, who entered participants' responses into a webform on a tablet. For a few sensitive questions relating to previous contraception use and sexual behaviour, participants entered their responses onto the tablet themselves. Participants were then remotely randomised into one of three arms. Intervention and contact arm participants received their first message the following day.

### Interventions
The interventions were categorised as per WHO classification of digital health interventions.[30] Arm 1 (intervention arm) was an on-demand information service to clients (WHO Classification 1.6). Participants received access to one domain of ARMADILLO content (eg, 'puberty and anatomy' or 'pregnancy prevention') each week and could request any 'subdomain' that interested them from the menu (eg, 'menstruation' or 'physical changes' for the puberty and anatomy domain or 'implants' or 'male condoms' for the pregnancy prevention domain) for free for the entire week (see online supplemental figure 1). Arm 2, (contact arm) employed targeted client communication (WHO Classification 1.1). Participants received the same system-initiated contacts as arm 1 participants but without access to the ARMADILLO content itself. Instead, a once-weekly SMS alerted them to an SRH domain for that week (eg, relationships, pregnancy, HIV) and encouraged them to learn on their own (see online supplemental figure 2). At the end of the week, participants in both arms were provided with a single SMS-based quiz question on that domain's content. If the participant answered (correctly or not), they received a small amount of airtime (1USD). Intervention and contact messages were available in either Swahili or English, per the participant's preference. All SMS costs were reverse-billed to the study, so intervention and contact arm participants incurred no costs from their participation. Those

## Box 1    Myths and misconceptions

⇒ Health—People who use contraceptives end up with health problems.
⇒ Body shape—Hormonal contraceptives are fattening.
⇒ Infertility—(1) After a woman uses contraceptive methods, it is difficult to get pregnant, and (2) use of a contraceptive injection can make a woman permanently infertile.
⇒ Harm—Contraceptives can harm a woman's womb.
⇒ Sex drive—Contraceptives reduce women's sexual urges.
⇒ Cancer—Contraceptives can cause cancer.
⇒ Malformations—Contraceptives can give you deformed babies.
⇒ Social constructs—(1) Birth control should be a female concern and (2) women who use family planning/birth-spacing may become promiscuous.

randomised to arm 3 (control arm) received standard of care (no messages).

### Assessments and outcome

The primary outcome was assessed using an index developed by the research team of 10 contraception myths and misconceptions (box 1). These were identified based on literature review and a series of focus group discussions with young people prior to the start of the RCT. In these sessions, young people used individual activities and group discussion centred around short vignettes of young couples thinking about starting contraception to describe local concerns around contraception use.[31] At baseline and endline, RCT participants were asked to state how much they agreed or disagreed with a given statement based on a four-point Likert scale.

### Sample size

The sample size was calculated such that it provided 80% power to detect a 10% change in mean number of myths believed from baseline to endline, assuming that baseline level of belief was 55%, type 1 error at 5% using two-sided Z test with continuity correction and unpooled variance and accounting for a dropout rate of up to 20%. The sample size accounted for Bonferroni correction due to three-arm pairwise comparisons. Based on the aforementioned, a minimum number of participants to be sampled was 705, split evenly across intervention, contact and control arms (1:1:1).

### Randomisation procedures

Participants were individually randomised to either intervention, contact or control group using a ratio of 1:1:1 as per computer-generated randomisation schedule developed using Node.js and docker. All the study participants had equal probability of being assigned to either arm. Allocation took place after the participant had completed the baseline survey. ARMADILLO was an open-label trial; however, neither the technological partner nor the research team had any control of arm assignments.

### Data analysis

The 10 items of the primary outcome were dichotomised from the original Likert scale (strongly agree, agree, disagree and strongly disagree) as follows: (1) agree and strongly agree (participant believed the myth—bad) were recoded as agree and coded as 1; (2) disagree and strongly disagree (participant rejected the myth—good) were recoded as disagree and coded as 0. A participant score for the 10 questions was generated with a total maximum score of 10, corresponding to the number of myths/misconceptions that the participant believed. The average number of myths/misconceptions believed by participants in each arm was computed. There were no missing values for the 10 items across all arms. The study participants responded to all the 10 primary outcome questions in the assessment.

The baseline factors were described using proportions. To ensure that oversampling in certain arms had no effect on the randomisation, we performed $\chi^2$ tests on demographic characteristics to confirm that there were no baseline differences between arms. To assess attrition bias (a systematic error caused by unequal loss of participants from the trial between the baseline and the endline), we used Fisher's exact $\chi^2$ tests for the sociodemographic variables to test whether participants lost to follow-up differed across the trial arms (online supplemental table 1) as well as if they differed from those who responded as a function of study group (online supplemental table 2).

First, we present proportions of those who believed in the myths at baseline and endline for all arms and the percentage change in the myths believed between the two periods. To obtain the average number of myths believed per participant, we computed (using the sum of the dichotomised 10-item response) the number of myths believed for each participant at baseline and at endline. Then for each participant, we computed the average number of myths believed (expressed as a percentage) at the baseline and at endline (by dividing the sum of myths believed by 10 and multiplying by 100). Next, for each participant, his or her absolute change in the average myths believed between the endline and baseline was computed (endline minus baseline).

Normality of the absolute changes in the myths believed was tested using the quantile-quantile plots. As the distribution of the absolute changes in the myths believed was normally distributed, analysis of variance (ANOVA) test of equality of group means was used to test the between group differences in the means of the absolute myths change. We estimated the difference-in-differences (DID) of the average number of myths believed by participants in a given arm to evaluate the effect of the ARMADILLO intervention to dispel myths and misconceptions about contraception. DID tells us whether the expected mean change in the number of myths and misconceptions believed from baseline to endline was different in the groups compared. DID is calculated by subtracting the average of the outcome in the control or contact arm from the average of the outcome in the intervention arm

$(d_1)$, where the outcome is the change in percentage number of myths believed by each individual between the endline and baseline. DID was also used to assess changes in the average proportion of myths believed per participant between the control and the contact arm $(d_2)$.

$$\hat{d_1} = \left\{ \left[ Mean\left(Y_i\left(Endline\right)\right) - Mean\left(Y_i\left(Baseline\right)\right) \right]_{Intervention} \right\} - \left\{ \left[ Mean\left(Y_i\left(Endline\right)\right) - Mean\left(Y_i\left(Baseline\right)\right) \right]_{j} \right\}$$

$$\hat{d_2} = \left\{ \left[ Mean\left(Y_i\left(Endline\right)\right) - Mean\left(Y_i\left(Baseline\right)\right) \right]_{Contact} \right\} - \left\{ \left[ Mean\left(Y_i\left(Endline\right)\right) - Mean\left(Y_i\left(Baseline\right)\right) \right]_{Control} \right\}$$

Where $i$ refer to the ith individual in the trial arm; while $j$=contact or control

All analyses were based on complete-case (CC) dataset while analyses based on per-protocol (PP) dataset were used for sensitivity analysis. Participants were included in the analysis provided that they had completed both baseline and endline surveys. In this case, the ITT analysis was equivalent to the CC.

Participants were included in the PP analysis provided that they had completed baseline and endline surveys, and that the intervention system could confirm that they had (1) received the ARMADILLO message domain associated with the primary outcome (pregnancy prevention); and (2) requested at least one message from this domain. Results with a type I error of p<0.05 in two-sided tests were considered statistically significant. Where Bonferroni correction was applied for the pairwise comparisons of the three study arms, p<0.017 were considered statistically significant. Analyses were performed using Stata V.15, and all were conducted in accordance with a prespecified statistical analysis plan.

## Patients and public involvement statement

ARMADILLO's population of 'young people' were involved in the study from its initial, formative stage,[28 32] which included message content development. They and the broader community continued engagement through this trial through the ARMADILLO community advisory board (CAB) consisting of representatives from the Ministries of Health, Education and Social Services; youth-led organisations; area chiefs; healthcare workers providing SRH services to adolescents; and young people themselves. CAB members provided technical and field support throughout the data collection period. Additionally, young people identified from the study area were trained as data collectors and hired to enumerate young people in the area as well as recruit and implement baseline and endline surveys with study participants. Young data collectors' input also shaped recruitment hours and strategies. A dissemination involving local and national stakeholders took place in July 2019—selected young data collectors participated in the dissemination meeting and shared their feedback.

## RESULTS

A total of 740 men and women aged 18–24 years were randomised into intervention, contact and control arms (figure 1).

In the intervention period which lasted 7 weeks, 116 of the 740 (16%) study participants dropped out over

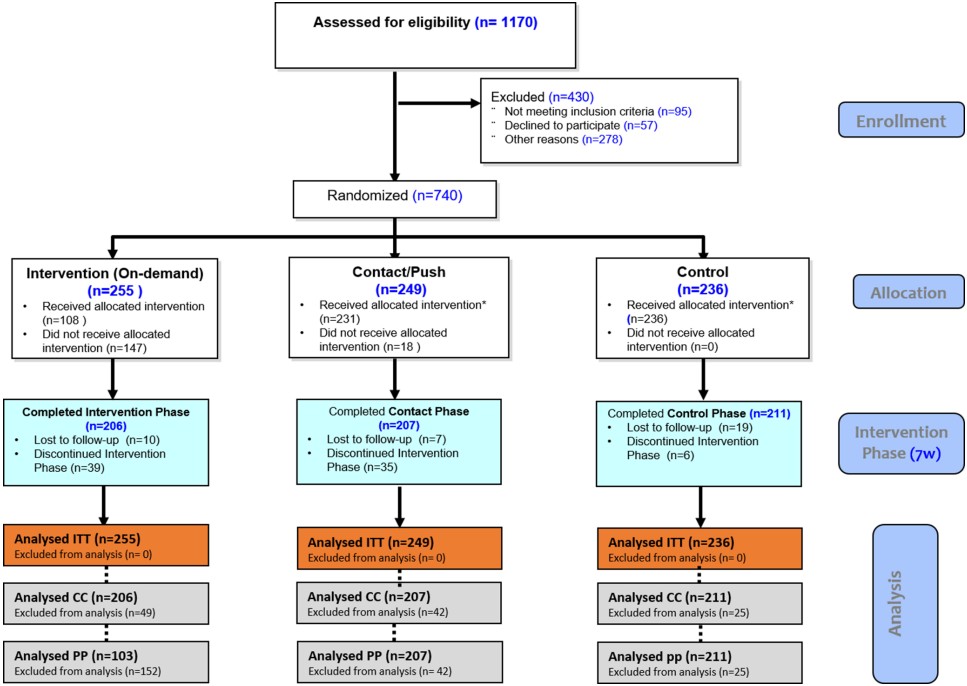

**Figure 1** ARMADILLO Kenya's Consolidated Standards of Reporting Trials diagram. ARMADILLO, Adolescent/Youth Reproductive Mobile Access and Delivery Initiative for Love and Life Outcomes; CC, complete case; PP, per protocol.

**Table 1** Baseline characteristics of the participants, by study arm (N=740)

| Characteristic | Intervention (I) N=255, n (%) | Contact (P) N=249, n (%) | Control (C) N=236, n (%) | Total N=740, n (%) |
|---|---|---|---|---|
| Age of the participant | | | | |
| 18–19 years | 62 (24.3) | 55 (22.09) | 53 (22.5) | 170 (23.0) |
| 20–24 years | 193 (75.7) | 194 (77.91) | 183 (77.5) | 570 (77.0) |
| Sex | | | | |
| Male | 134 (52.6) | 133 (53.41) | 126 (53.4) | 393 (53.1) |
| Female | 121 (47.5) | 116 (46.59) | 110 (46.6) | 347 (46.9) |
| Education level | | | | |
| Never gone to school | 9 (3.5) | 7 (2.81) | 12 (5.1) | 28 (3.8) |
| Primary school | 92 (36.1) | 97 (38.96) | 80 (33.9) | 269 (36.4) |
| Secondary school | 117 (45.9) | 118 (47.39) | 119 (50.4) | 354 (47.8) |
| Postsecondary education | 37 (14.5) | 27 (10.84) | 25 (10.6) | 89 (12.1) |
| Sublocation | | | | |
| Ngombeni | 34 (13.3) | 38 (15.26) | 43 (18.2) | 115 (15.5) |
| Kitivo | 9 (3.5) | 9 (3.61) | 9 (3.8) | 27 (3.7) |
| Simkumbe | 20 (7.8) | 17 (6.83) | 21 (8.9) | 58 (7.8) |
| Mkoyo | 8 (3.1) | 8 (3.21) | 8 (3.4) | 24 (3.2) |
| Gombato | 17 (6.7) | 11 (4.42) | 9 (3.8) | 37 (5.0) |
| Ukunda | 167 (65.5) | 166 (66.67) | 146 (61.9) | 479 (64.7) |
| Person currently living with | | | | |
| Living alone | 24 (9.4) | 21 (8.43) | 21 (8.9) | 66 (8.9) |
| Living with others | 231 (90.6) | 228 (91.57) | 215 (91.1) | 674 (91.1) |
| Current relationship status | | | | |
| Single | 128 (50.2) | 121 (48.59) | 118 (50.0) | 367 (49.6) |
| Friends with benefits/dating/cohabiting/engaged | 109 (42.8) | 104 (41.77) | 92 (39.0) | 305 (41.2) |
| Married | 18 (7.1) | 24 (9.64) | 26 (11.0) | 68 (9.2) |
| Parity | | | | |
| None | 224 (87.8) | 211 (84.7) | 191 (80.9) | 626 (84.6) |
| One child | 24 (9.4) | 28 (11.2) | 35 (14.8) | 87 (11.8) |
| 2+ children | 7 (2.8) | 10 (4.0) | 10 (4.2) | 27 (3.7) |
| First birth age | | | | |
| Never given birth | 224 (87.8) | 211 (84.7) | 191 (80.9) | 626 (84.6) |
| ≤19 years (adolescents) | 18 (7.1) | 22 (8.8) | 23 (9.8) | 63 (8.5) |
| ≥20 years (young women) | 13 (5.1) | 16 (6.4) | 18 (7.6) | 47 (6.4) |

the course of the 7-week intervention period (intervention arm—49 (19%); contact arm—42 (17%); control arm—25 (11%)). Among participants in the intervention arm, 206 (81%) completed both the baseline and the endline assessments (making them eligible for CC analysis) while 103 (40%) were eligible for PP analysis. Among the contact group, 207 (83%) received the push messages and were included in both the CC and PP analysis. In the control group, 211 (89.4%) completed both the baseline and the endline assessments. Baseline characteristics for the study sample are shown in table 1. Across all arms, 53% of the participants were male, 48% had a secondary

education or higher, 65% were from Ukunda sublocation, 91% lived with others and 85% did not have children. There were no significant baseline differences between the intervention, contact and control groups.

Concerning attrition bias, the analysis revealed that participants who dropped out in each arm were similar to each other (online supplemental table 1). However, there was a significant association between dropping out of the study and the number of children the participant had at the time of the study in the control and the contact groups' participants, online supplemental table 1. The analysis assessing attrition bias also revealed that there

**Table 2** Intervention effects for dichotomous outcomes—complete-case analysis

| Myth | Control (n=211) | | | Intervention (n=206) | | | Contact (n=207) | | |
|---|---|---|---|---|---|---|---|---|---|
| | *Baseline | *Endline | Diff. | *Baseline | *Endline | Diff. | *Baseline | *Endline | Diff. |
| Population based analysis | | | | | | | | | |
| Hormonal contraceptives are fattening | 139 (65.9%) | 117 (55.5%) | −10.4% | 144 (69.9%) | 135 (65.5%) | −4.4% | 142 (68.6%) | 127 (61.4%) | −7.2% |
| Contraceptives can harm a woman's womb | 135 (64%) | 114 (54%) | −10% | 125 (60.7%) | 106 (51.5%) | −9.2% | 144 (69.6%) | 112 (54.1%) | −15.5% |
| People who use contraceptives end up with health problems | 133 (63%) | 103 (48.8%) | −14.2% | 120 (58.3%) | 105 (51%) | −7.3% | 135 (65.2%) | 100 (48.3%) | −16.9% |
| Contraceptives can cause cancer | 117 (55.5%) | 88 (41.7%) | −13.8% | 112 (54.4%) | 90 (43.7%) | −10.7% | 123 (59.4%) | 86 (41.6%) | −17.8% |
| Use of a contraceptive injection can make a woman permanently infertile | 108 (51.2%) | 82 (38.9%) | −12.3% | 102 (49.5%) | 73 (35.4%) | −14.1% | 123 (59.4%) | 77 (37.2%) | −22.2% |
| Contraceptives reduce women's sexual urges | 101 (47.9%) | 65 (30.8%) | −17.1% | 83 (40.3%) | 57 (27.7%) | −12.6% | 107 (51.7%) | 76 (36.7%) | −15% |
| Contraceptives can give you deformed babies | 98 (46.5%) | 69 (32.7%) | −13.8% | 92 (44.7%) | 53 (25.7%) | −19% | 106 (51.2%) | 71 (34.3%) | −16.9% |
| After a woman uses contraceptive methods, it is difficult to get pregnant | 92 (43.6%) | 81 (38.4%) | −5.2% | 101 (49%) | 69 (33.5%) | −15.5% | 102 (49.3%) | 80 (38.7%) | −10.6% |
| Birth control should be a female concern | 59 (28%) | 41 (19.4%) | −8.6% | 46 (22.3%) | 32 (15.5%) | −6.8% | 59 (28.5%) | 35 (16.9%) | −11.6% |
| Women who use family planning/birth-spacing may become promiscuous | 109 (51.7%) | 92 (43.6%) | −8.1% | 108 (52.4%) | 84 (40.8%) | −11.6% | 111 (53.6%) | 89 (43%) | −10.6% |
| Subject specific analysis | | | | | | | | | |
| Average # myths believed, per participant (SE) | **5.17** | **4.04** | | **5.01** | **3.90** | | **5.57** | **4.12** | |
| Average absolute change in myths believed | **−1.13** | | | **−1.11** | | | **−1.44** | | |
| CI of the diff. | **(−1.59 to −0.67)** | | | **(−1.54 to −0.68)** | | | **(−1.91 to −0.98)** | | |
| Percentage absolute change in myths believed | **−21.9** | | | **−22.2** | | | **−25.9** | | |
| Test of diff. in the mean of the absolute change in myths believed | Control (n=211) | | | Intervention (n=206) | | | Contact (n=207) | | |
| ANOVA test F statistics | **0.66** | | | | | | | | |
| P value | **0.5181** | | | | | | | | |

n*−Those who believed the myth (had wrong answer).

Numbers in bold are for variables analysed under subject specific analysis

ANOVA, analysis of variance.

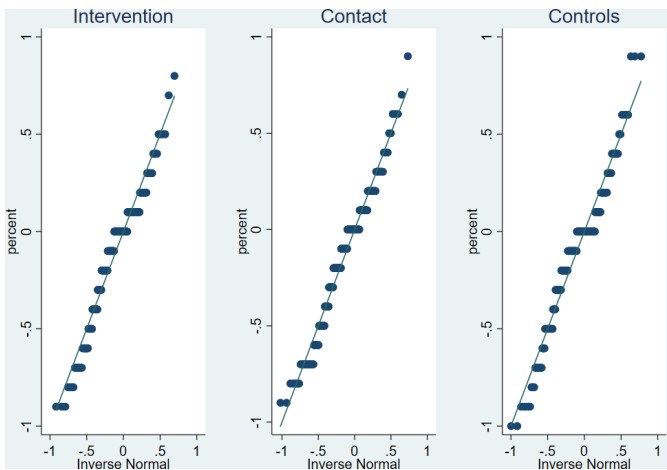

**Figure 2** Checking the normality of the absolute changes in the myths believed.

was no significant difference in the sociodemographics between the participants who were lost to follow-up and those who finished the 7 weeks of the intervention and took the endline survey, online supplemental table 2.

The results of the CC analysis examining the myths and misconceptions believed are displayed in table 2. The myths are ordered by those which were believed by the most number of participants, across groups, with the most salient myth at the top. The results show that at baseline, study participants in all arms believed around half of the myths related to contraception on average. At the end of the 7-week intervention period, the average number of myths and misconceptions believed per participant had significantly decreased for all the three groups (p<0.0001). The average absolute decrease in the myths believed was 11.1% among the intervention group (mean −11.1%; 95% CI −17.1% to −5.2%); 14.4% among the contact group (mean −14.4%; 95% CI −20.5% to −8.4%) and 11.3% among the control group (mean −11.3%; 95% CI −17.4% to −5.2%). From the normality test shown in figure 2, the absolute change in all the three groups were normally distributed. The ANOVA test of equality-of-populations means showed that there was no significant difference in the group medians (p=0.5181).

As presented in the DID analysis in table 3, there was no statistically significant differences between the baseline to endline decrease observed across the three arms (p>0.017-Bonferroni corrected significance level).

Effective sample size used for the analysis was 206 participants for the intervention group; 207 participants for the contact group and 211 participants for the control group.

Sensitivity analysis including only participants who met PP inclusion criteria did not alter the findings from CC analysis. As the results of the CC analysis, and PP analysis did not differ with respect to statistical significance within groups or between groups, only those for the CC analysis are reported (PP analysis is attached as online supplemental table 3).

Finally, table 4 shows estimates of a possible source of contamination between study arms. Of the 23% of the participants in the intervention group who shared the messages with others, 13%, 5% and 4% shared the messages with friends, partners and multiple contacts, respectively. Among the 27% contact group participants who shared the messages with others, 15%, 8%, 1% and 3% shared the messages with friends, partners, siblings and multiple contacts, respectively.

## DISCUSSION

Our findings suggest that provision of SRH content via SMS is potentially useful in dealing with contraception-related myths and misconceptions among youth. Across arms, the study demonstrated between a 11% and 14% reduction in the average number of myths/misconception statements believed over the study period. However, we did not observe a significant difference in the magnitude of reduction between the arms. Therefore, despite the significant decrease in myths-believed that we observed between baseline and endline, we are unable to conclusively state that the ARMADILLO intervention was better than SMS 'contact' or no intervention at all.

One possible reason for not seeing a significant effect of the ARMADILLO intervention versus no intervention is that correcting false information is difficult. Studies aimed at correcting misinformation about vaccines, for

| Table 3 | Difference in difference analysis | | |
|---|---|---|---|
| **Outcome** | **Percentage point differences (95% CI)** | | **P value** |
| Contraception myths and misconceptions index score (endline – baseline assessment) | | | |
| Arm 1: Intervention (MeanΔ, 95% CI) | −11.1% (−17.1% to −5.2%) | | <0.001 |
| Arm 2: Contact (MeanΔ, 95% CI) | −14.4% (−20.5% to −8.4%) | | <0.001 |
| Arm 3: Control (MeanΔ, 95% CI) | −11.3% (−17.4% to −5.2%) | | <0.001 |
| Mean (Δ Intervention) – Mean (Δ Control) | 0.2% (−8.3% to 8.7%) | | 0.961 |
| Mean (Δ Contact) – Mean (Δ Control) | −3.1% (−11.7% to 5.4%) | | 0.475 |
| Mean (Δ Intervention) – Mean (Δ Contact) | 3.3% (−5.1% to 11.8%) | | 0.440 |

Δ refers to the subject-specific change in the outcome from baseline to endline. 95% CI refers to the 95% CI. A generalised linear model using a normal distribution and identity link was used to compare scores.

**Table 4** Study contamination

| | | Shared the messages | Shared the messages with: | | | | |
|---|---|---|---|---|---|---|---|
| | | | Siblings | Friends | Partner | Parents | Multiple contacts |
| Intervention (N=206) | n | 47 | 0 | 26 | 10 | 2 | 9 |
| | % | 22.8 | 0.0 | 12.6 | 4.9 | 1.0 | 4.4 |
| Contact (N=207) | n | 55 | 2 | 30 | 16 | 0 | 7 |
| | % | 26.6 | 1.0 | 14.5 | 7.7 | 0.0 | 3.4 |
| Total (N=413) | n | 102 | 2 | 56 | 26 | 2 | 16 |
| | % | 24.7 | 0.5% | 13.6% | 6.3% | 0.5% | 3.9% |

example, have shown that even when attempts to correct invalid information do not entrench the original misinformation, they can frequently fail because people cannot successfully update their memories, and still fall back on information they know is not correct.[33–36] A 7-week digital health intervention, dedicated to SRH broadly, may not have been enough to dispel deeply entrenched concerns about contraception. Myths and misconceptions around contraception are also particularly tricky given that misconception about contraception generally (eg, that contraception use can lead to infertility) may be partially rooted in individual experiences of real side effects (eg, the possible delay of a return to fertility following use of injectable contraception).[12]

Alternatively, we may have seen no difference between the arms because the intervention was truly not better than SMS-prompted self-learning and/or no intervention at all. Several other RCTs that have attempted to tie adolescent-targeted digital health interventions to SRH knowledge, acceptability or behavioural outcomes have resulted in similarly inconclusive findings,[26 37–40] indicating that digital interventions on their own may not be enough to encourage behavioural change. However, while the above reasons would explain the lack of difference between arms, they do not explain why participants in all arms believed significantly fewer myths at endline than they did at baseline.

Here, contamination between arms may be to blame. RCT of digital health client communication interventions are difficult, especially when the intervention content can be easily shared between neighbours and across communities. To the best of our knowledge, there were no other health campaigns or interventions aimed at dispelling the myths and misconceptions in the area during the study period. However, about a quarter of the intervention and contact arm participants reported sharing information with study participants and other members of the community. Any effect of participants sharing information was amplified by our participants being randomised at the individual level. This resulted in participants in different arms living in close proximity, often in neighbouring households. ARMADILLO intervention arm participants may therefore have received the messages and shared them with their friends/neighbours, some of whom were ARMADILLO contact and control arm participants.

Addressing/dispelling myths and misconceptions among youth is particularly important for future contraception use. By nature of originating in social networks as well as their likelihood to 'stick' indefinitely, myths and misconceptions among youth should be dispelled early to prevent their becoming engrained.[6 41] Indeed, our study found that two of the three most commonly-believed myths and misconceptions among youth age 18–24 years (people who use contraceptives end up with health problems, and contraceptives can harm a woman's womb) were also the top myths for youth and adult women aged 15–49 years in Kenya, Nigeria and Senegal.[9]

Other studies have reported successfully addressing myths and misconceptions with dedicated community and communication interventions,[42] involving a variety of opinion leaders and channels.[9 42] Following a 4-year intervention using radio, religious leaders and community health volunteers, for example, Kenya's Tupange study reported a 15% decrease in the number of women who believed myths and misconceptions statements between baseline and endline.[42]

Digital interventions are an additional channel which can be included in this mix. There is a wealth of opportunity to engage with young people en masse not only through SMS and voice channels, but also by widely used messaging and social media platforms like WhatsApp, Facebook and Facebook Messenger, the latter of which don't have the bulk telecom-related costs of SMS and voice interventions. The popularity of the ARMADILLO interventions among its users—one silver lining of the contamination between arms—indicate that such interventions can be considered as one tool in a multipronged approach targeting young people with correct SRH information.[25 43–45] However, evidence continues to be needed on adolescent-targeted client communication interventions in general.

This study is not without limitations. The lack of differences between the intervention and the other arms could very well be due to our adoption of individually randomised rather than cluster randomised study design. A cluster design was considered; however, the accessibility and make-up of the study area did not allow for homogeneous clusters to be created (Ukunda, eg, is unique in Kwale County for its population density). While unfortunate for the results of the trial, it demonstrates that the

messages were as popular as the research team hoped and provides some positive insight as to the dissemination which might take place if a similar intervention is implemented outside of a research setting. The study is strengthened by its choice of primary outcome: we intentionally avoided powering the study around SRH behavioural outcomes, building on learnings from previous studies that light-touch digital interventions alone may not be enough to see behavioural change in such a complex area of health.

In conclusion, creative and consistent interventions are needed to address deeply rooted myths and misconceptions young people have about contraception and mitigate one important driver of anxiety around contraception use in Kenya. These can include digital interventions. However, while the ARMADILLO study saw some promising results, we cannot conclusively say that digital interventions alone are sufficient to affect change in SRH-related outcomes. Additional research (using alternative designs) will be required to identify the specific value of digital targeted client communication programmes in addressing young people's contraception myths/misconceptions and improving SRH knowledge overall.

**Author affiliations**
[1]Administration, Technical University of Mombasa, Mombasa, Kenya
[2]4-PSRI, International Centre for Reproductive Health Kenya, Mombasa, Kenya
[3]Department of Sexual and Reproductive Health and Research including UNDP/UNFPA/UNICEF/WHO/World Bank Special Programme of Research, Development and Research Training in Human Reproduction (HRP), World Health Organization, Geneve, Switzerland
[4]4-PSRI, University of Nairobi, Nairobi, Kenya
[5]4-PSRI, National Council for Population and Development, Nairobi, Kenya
[6]4-PSRI, Aga Khan University - Kenya, Nairobi, Kenya
[7]4-PSRI, International Centre for Reproductive Health, Ghent, Oost-Vlaanderen, Belgium

**Correction notice** This article has been corrected since it was first published. The authorship information has been updated.

**Acknowledgements** The authors thank the youth in Kwale who participated in the study, the research assistants who collected data, and Winnie Wangari who coordinated and supervised data collection. The authors are grateful for feedback given by Carol Mukira and Nelly Ibeere.

**Contributors** PG and LG contributed equally to this paper (in study design, interpretation of data and writing of the manuscript). JM, MT, NH, MW and LS contributed to the design of the ARMADILLO study. JM contributed to data collection; MW and MT contributed to the analysis; PG, LG, JM, MT, NH, HS, AA, MT and LS contributed to interpretation of data. PG and LG wrote the first draft of the paper, and all authors contributed to its revision; all authors approve of the submitted article. The manuscript represents the views of the named authors only. PG acts as a gaurantor.

**Funding** This work was funded by the UNDP/UNFPA/UNICEF/WHO/World Bank Special Programme of Research, Development and Research Training in Human Reproduction (HRP).

**Competing interests** None declared.

**Patient consent for publication** Not applicable.

**Ethics approval** The study was approved by the ethics committees at the Kenyatta National Hospital (KNH ERC P550/09/2014) and the WHO (Protocol WHO A65892).

**Provenance and peer review** Not commissioned; externally peer reviewed.

**Data availability statement** Data are available on reasonable request. De-identified data related to this article can be obtained by reasonable request to the corresponding author (ORCID: 0000-0001-9636-165X).

**ORCID iDs**
Lianne Gonsalves http://orcid.org/0000-0003-2409-5043
Michael Waithaka http://orcid.org/0000-0002-6766-9492

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
