## [Reviewer comments · BMJ Open]

ARTICLE DETAILS

TITLE (PROVISIONAL)	Busting contraception myths and misconceptions among youth in Kwale County, Kenya: Results of a digital health randomized control trial
AUTHORS	Gichangi, Peter ; Gonsalves, Lianne; Mwaisaka, Jefferson; Thiongo, Mary; Habib, Ndema; Waithaka, Michael; Tamrat, Tigest; Agwanda, Alfred; Sidha, Hellen; Temmerman, Marleen; Say, Lale

VERSION 1 – REVIEW

REVIEWER	Smith, Chris London School of Hygiene & Tropical Medicine, Clinical Research Department
REVIEW RETURNED	17-Jan-2021

GENERAL COMMENTS	1. Title: 'Bursting' or 'Busting'?2. Please reconsider use of the terms 'myths and misconceptions'. Side-effects experienced from contraception methods can be real and not always be attributed to myths and misconceptions. Therefore please consider clarification regarding what you mean by a myth and a misconception and how this differs from a real experienced side-effect such as menstrual changes, feeling of body heat, skin changes, delayed return to fertility (with injectable) or rare serious health events (such as thromboembolism). This comment applies to several sections of the manuscript e.g. introduction (para 2), box 1 (please define what is meant by 'health problems?'), and discussion. Sorry, this is just my opinion but may not be shared by others and happy for the authors to push back on this. Some papers have discussed these issues e.g. Side effects and the need for secrecy: characterising discontinuation of modern contraception and its causes in Ethiopia using mixed methods and Menstrual Bleeding Changes Are NORMAL: Proposed Counselling Tool to Address Common Reasons for Non-Use and Discontinuation of Contraception.3. Few similar studies of digital health interventions for contraception are cited apart from McCarthy et al. Most studies in the background are about STIs/HIV yet the intervention is about contraception. Quite a few studies exist so please consider include some. And then discuss findings of your studies against those studies in the discussion.
--

	4. I cannot comment on the statistical methods (someone more qualified could review) but good to state if the statistical analysis plan was pre-specified or not (sorry if I missed that)
--	---

REVIEWER	Weis, Julianne USAID Office of Population and Reproductive Health
REVIEW RETURNED	06-May-2021

GENERAL COMMENTS	Good paper, contributes to growing evidence of mixed to null results of digital interventions in SRH knowledge of youth.
--

REVIEWER	Sheikhansari, Narges University of Exeter, Medical School
REVIEW RETURNED	11-Aug-2021

GENERAL COMMENTS	Thank you for this very interesting study.
--

VERSION 1 – AUTHOR RESPONSE

Reviewer: 1

Dr. Chris Smith, London School of Hygiene & Tropical Medicine, Nagasaki University Comments to the Author:

1. Title: 'Bursting' or 'Busting'?

Thank you for this – there was discussion of both in the team originally, with this feedback we have modified to 'busting'.

2. Please reconsider use of the terms 'myths and misconceptions'. Side-effects experienced from contraception methods can be real and not always be attributed to myths and misconceptions. Therefore please consider clarification regarding what you mean by a myth and a misconception and how this differs from a real experienced side-effect such as menstrual changes, feeling of body heat, skin changes, delayed return to fertility (with injectable) or rare serious health events (such as thromboembolism). This comment applies to several sections of the manuscript e.g. introduction (para 2), box 1 (please define what is meant by 'health problems?'), and discussion. Sorry, this is just my opinion but may not be shared by others and happy for the authors to push back on this. Some papers have discussed these issues e.g. Side effects and the need for secrecy: characterising discontinuation of modern contraception and its causes in Ethiopia using mixed methods and Menstrual Bleeding Changes Are NORMAL: Proposed Counselling Tool to Address Common Reasons for Non-Use and Discontinuation of Contraception.

We very much appreciate the reviewer's point here and agree fully with their point about real side effects often being blended with broader myths/misconception etc. As requested, we have endeavoured to clarify how we define myths and misconceptions in the introduction, relying on this PATH brief from 2015 (now referenced in the paper). We have also added an additional sentence describing how the Box 1 myths/misconceptions were identified to the Methods section and cite a paper from the formative research for readers to get additional information. Finally, as requested, we acknowledge the overlap between experienced side effects and broader misconceptions in the Discussion as well.

For the reviewer's ease and interest, we provide additional information below:

In our own identification of this study's index of 10 myths and misconceptions, we started with a list of region-specific myths/misconceptions (identified from the literature) and then refined these heavily based on formative work done in the study area prior to the start of the RCT. Here, young men and women were asked to reflect on some short stories featuring couples their age who were considering starting

contraception but were 'nervous about what they had heard from friends'. They were then asked to share what these couples may have heard – this reflects the broad nature (e.g. the 'health problems') of one in the list.

The study group refined the list to feature those identified as the most salient based on FGDs. We acknowledge that reductions in libido (specific to certain methods like injectables) and injectable-specific return to fertility delay that the reviewer mentions can still represent actual side effects for some women for some methods. We contend that the extrapolation of these method-specific experiences to contraception in general is part of what turns these into persistent and widespread misconceptions about **all** contraception. An unfortunate result is the high discontinuation of contraception altogether because of side-effects experienced with one method, rather than seeking to find a better fit.

3. Few similar studies of digital health interventions for contraception are cited apart from McCarthy et al. Most studies in the background are about STIs/HIV yet the intervention is about contraception. Quite a few studies exist so please consider include some. And then discuss findings of your studies against those studies in the discussion.

We have added references to two additional papers in the introduction but note that for the introductory paragraph we have endeavoured to reflect the real impetus that resulted in the ARMADILLO study's creation – the availability of thin but promising data in relevant areas of health, but none at the time specific to digital health interventions targeted at improving contraception outcomes among young people.

We have also added additional, more recent citations to the third paragraph in the discussion where we note our null findings (including results from two more recent trials in Tajikistan and Bolivia from McCarthy and colleagues). However we have done very minimal text modification here as these studies also had null findings for outcomes relevant to this study. In line with the comment from reviewer 2 below, we agree this reflects the general mixed to null findings from this space, which may be due to recruitment and implementation difficulties around implementing RCTs for digital health interventions (we are up front about our own challenges with contamination in the Discussion), but also that these may not be the most appropriate, stand-alone interventions for improving contraception-related outcomes among young people (an example of digital health enthusiasm having outpaced evidence).

4. I cannot comment on the statistical methods (someone more qualified could review) but good to state if the statistical analysis plan was pre-specified or not (sorry if I missed that)

Thank you for noticing this omission – the analysis plan was indeed pre-specified and we had failed to indicate this. We now do so in the last sentence of the 'datanalysis' subsection. 'Analyses were performed using Stata version 15, and all were conducted in accordance with a prespecified Statistical Analysis Plan.'

Reviewer: 2

Dr. Julianne Weis, USAID Office of Population and Reproductive Health Comments to the Author: Good paper, contributes to growing evidence of mixed to null results of digital interventions in SRH knowledge of youth.

Our thanks for taking the time to review this paper and for the kind words.

Reviewer: 3

Dr. Narges Sheikhsari, University of Exeter Comments to the Author: Thank you for this very interesting study.

Our thanks for taking the time to review this paper.

Reviewer: 1

Competing interests of Reviewer: None declared

Reviewer: 2

Competing interests of Reviewer: None

Reviewer: 3

Competing interests of Reviewer: None

VERSION 2 – REVIEW

REVIEWER	Smith, Chris London School of Hygiene & Tropical Medicine, Clinical Research Department
REVIEW RETURNED	30-Sep-2021
GENERAL COMMENTS	Many thanks to the authors for their responses. I do not have any further comments and recommend publication of the study